# Discrimination and Characterization of Volatile Flavor Compounds in Fresh Oriental Melon after Forchlorfenuron Application Using Electronic Nose (E-Nose) and Headspace-Gas Chromatography-Ion Mobility Spectrometry (HS-GC-IMS)

**DOI:** 10.3390/foods12061272

**Published:** 2023-03-16

**Authors:** Qi Wang, Xueying Chen, Chen Zhang, Xiaohui Li, Ning Yue, Hua Shao, Jing Wang, Fen Jin

**Affiliations:** Key Laboratory of Agro-Product Quality and Safety, Institute of Quality Standards & Testing Technology for Agro-Products, Chinese Academy of Agricultural Sciences, Beijing 100081, China; wangqi2021yw@163.com (Q.W.); chenxueying0327@163.com (X.C.); lxhui33362@163.com (X.L.);

**Keywords:** oriental melon, volatile compounds, forchlorfenuron, flavor, headspace-gas chromatography-ion mobility spectrometry, electronic nose

## Abstract

Aroma is a crucial factor determining the market value and consumer satisfaction of fresh oriental melon. However, few studies focus on the volatile flavor of fresh oriental melon, and the effect of forchlorfenuron application on the aroma profile is unclear. This study characterized the volatile profile of fresh oriental melon fruit after forchlorfenuron application by E-nose and HS-GC-IMS. The holistic variation of volatile compounds exhibited evident distinction based on linear discriminant analysis (LDA) with E-nose. Forty-eight volatile compounds were identified from fresh oriental melon via GC-IMS, mainly esters, alcohols, aldehydes, and ketones, along with smaller quantities of sulfides and terpenes. Compared to pollination melon fruits, 13 critical different volatile flavor compounds were screened out in forchlorfenuron application groups by the PLS-DA model, imparting sweet fruity flavor. The results of the current study provide a valuable basis for evaluating the flavor quality of oriental melon after forchlorfenuron treatment.

## 1. Introduction

Oriental melon is a species of thin-pericarp melon [1], and it has the largest plantation in China, accounting for about 51% of the total global production. Oriental melons are often planted in greenhouses to increase the price of fruit. However, the lack of pollinators often affects the fruit-set rate for facilities. Forchlorfenuron is a synthetic cytokinin-like growth regulator, which can act synergistically with endogenous auxins to induce parthenocarpy and promote cell expansion [2]. In recent years, forchlorfenuron has been extensively used in oriental melon cultivation to improve the fruit set.

With the increasing prevalence of forchlorfenuron application, more and more studies have focused on its influence on fruit quality [3,4]. Several studies have shown that the application of forchlorfenuron decreased sucrose and glucose content and increased bitterness in melon [5,6]. In addition to sugar, volatile aroma plays a decisive role in the purchase of oriental melon [7]. The volatile components of melon have been analyzed in previous reports, and approximately 300 compounds have been identified [8,9,10,11]. They produce volatile aldehydes, alcohols, and especially large quantities of esters, likely to be the key contributors to their unique aroma [12,13,14]. However, limited studies have reported the effect of forchlorfenuron on the aroma compounds of oriental melon fruit. Although Li et al. found that the abundance of volatile compounds was decreased after forchlorfenuron application in muskmelon using gas chromatography–mass spectrometry,^13^ the findings have generally been obtained from frozen samples as an alternative to fresh samples. It has been reported that significant changes in volatiles occurred during the freezing process in fruits and vegetables [15,16,17]. To accurately evaluate the effect of forchlorfenuron on aroma characteristics, a quick and straightforward method to discriminate the variation of aroma volatiles using fresh oriental melon fruit is critical.

Sensory analysis using trained panelists has been employed conventionally to evaluate the variation of fruit aroma, which can directly measure the fruit flavor intensity. However, this method is expensive and time-consuming, with low objectivity and reproducibility [18,19]. Electronic nose (E-nose) and gas chromatography–ion mobility spectrometry (GC-IMS), as emerging techniques for volatile-compound analysis, offer advantages of fast detection speed, high sensitivity, and little sample pretreatment [20,21,22,23,24]. These techniques have been successfully utilized individually or in combination in many fields, mainly involving freshness prediction, adulteration identification, and food composition classification. Ezhilan et al. discriminated the pathogen contamination of apples using E-nose; higher classification accuracy was attained with an accuracy of 99.9% [25]. The Guo et al. study showed the potential of GC-IMS-based approaches to evaluate the volatile compound profiles of fresh-cut yam at different stages in the yellowing period [26].

Therefore, the present study aims to identify the differentiation of flavor changes in oriental melon treated with different concentrations of forchlorfenuron. E-nose and HS-GC-IMS were applied to characterize the volatile compound composition and content when the oriental melon fruits were harvested after maturation. The results will provide new theoretical guidance for the more appropriate use of forchlorfenuron in oriental melon.

## 2. Materials and Methods

### 2.1. The Oriental Melon Field Trials

Oriental melons (*Cucumis melo* var. *makuwa*) were cultivated in a greenhouse during a summer–autumn cycle with common growing conditions. The temperature of the greenhouse was maintained in the range of 25–30 °C, with 60% average relative humidity throughout the experiment. The oriental melons were divided into three forchlorfenuron application groups and one pollination group. In the treatment groups, doses of forchlorfenuron soluble concentrate (SL) were set from 10 to 20 mg/L according to the recommended dose on the registered label, and the melon ovary was completely dipped with forchlorfenuron solutions (10 mg/L, 15 mg/L, 20 mg/L) for 1–2 s, respectively. In the pollination group, only the chasmogamy of female flowers by male flowers was considered. All fruit-set treatments were performed on the same morning (6–9 AM).

Representative melon fruit samples were harvested with the best edible quality according to the experience of melon farmers (34 days after pollination or forchlorfenuron application). Mature melon fruits were selected using a combination of different harvest indices, including smooth-skinned with sweet and fragrant pulp, aroma emission detected by the human nose, pale yellow skin color, and peduncle suberization (Appendix A) [11]. In addition, melons were selected in this experiment based on uniform size, weight, and color, and at least six fruits were collected in each group. Melon samples were placed in polyethylene bags and transported to the laboratory for the next stage.

### 2.2. Sample Preparation

The samples were hand cut with a sharp knife into 2 cm slices, from which the blossom ends and the stem were discarded. Tissues were immediately smashed by a pulverizer, and melon samples were placed into headspace vials, sealed until analysis.

### 2.3. E-Nose Analysis

The volatile profile of fresh oriental melon fruit was detected by PEN 3 E-nose (AIRSENSE Company, Schwerin, Germany). The E-nose consists of ten different metal oxide sensors. Each sensor has its corresponding sensitive substances: sensor 1 W1C is sensitive to aromatic compounds; sensor 2 W5S is sensitive to oxynitride; sensor 3 W3C is sensitive to ammonia and aromatic compounds; sensor 4 W6S is sensitive to hydrogen; sensor 5 W5C is sensitive to alkanes and aromatic compounds; sensor 6 W1S is sensitive to methane; sensor 7 W1W is sensitive to sulfur compounds; sensor 8 W2S is sensitive to ethanol; sensor 9 W2W is sensitive to aromatic and organic sulfur compounds; and sensor 10 W3S is sensitive to long-chain alkanes [27,28]. Samples (10 g) were placed in a 100 mL beaker and sealed with tin foil for 60 min. The determination conditions were as follows: the flow rate of carrier gas (pure dry air) was 400 mL/min, pre-injection time was 5 s, sample measurement time was 100 s, reset time was 5 s, and cleaning time was 100 s.

### 2.4. HS-GC-IMS Analysis

The identification of the characteristic volatile compounds of fresh oriental melon fruit was performed using a FlavourSpec^®^ ion mobility spectrometry (IMS) instrument (G.A.S., Dortmund, Germany) equipped with an auto-sampler unit, a syringe, a heated splitless injector, and a radioactive ionization source for headspace (HS) analysis.

The detection processes of HS-GC-IMS were conducted as described by Guo et al. [26] and adjusted slightly according to fresh oriental melon fruit characteristics. A homogenized oriental melon sample (2 g) was transferred to a 20 mL headspace bottle and was incubated at 40 °C for 20 min. Then, 200 μL was sampled from the headspace and was automatically injected into the heated injector at a temperature of 45 °C. After injection, GC was performed with a 15 m standard capillary column (FS-SE-54-CB capillary column, 15 m × 0.53 mm) to separate volatile compounds. The flow of carrier gas (nitrogen gas, 99.99% purity) was set at 2.0 mL/min. The analytes were ionized by a tritium source (6.5 keV) at atmospheric pressure and then transferred to the drift tube (98 mm length). Four groups of melon samples were detected in sequence by GC-IMS (repeated four times), which takes 25 min per sample. The retention index (RI) of volatile compounds was calculated using standardized n-ketones (Sinopharm Chemical Reagent Beijing Co., Ltd., Beijing, China) whose RI was linear. Compounds were identified by comparing RI and drift time (Dt, the time required for ions to reach the collector through the drift tube, in milliseconds) to the standard in the GC × IMS Library supplied by G.A.S. (Dortmund, Germany). The GC-IMS fingerprint analysis was conducted by comparing GC retention time and IMS drift time.

### 2.5. Multivariate Analysis

Linear discriminant analysis (LDA) was performed using the E-nose software system. The instrumental analysis software includes the Laboratory Analytical Viewer (LAV, G.A.S., Dortmund, Germany) and three plug-ins, as well as the GC × IMS Library Search, which can be used for sample analysis from different angles.

OriginPro 9.1 (Origin Lab Corporation, Northampton, MA, USA) was used to draw the radar chart. Principal component analysis (PCA) and partial least squares discriminant analysis (PLS-DA) were conducted using SIMCA-P software v14.1 (Umetrics, Umea, Sweden).

The odor type and odor strength were obtained from The Good Scents Company Information System. (http://www.thegoodscentscompany.com/index.html (accessed on 9 March 2023)).

## 3. Results and Discussion

### 3.1. Evaluation of the Volatile Compounds of Fresh Oriental Melon Fruit by E-Nose

E-nose analysis was conducted on fresh oriental melon from different treatment groups to monitor shifts in aroma composition. The odor radar map of volatile compounds in fresh oriental melon is presented in Figure 1A, following detection using 10 odor sensors. It was found that the W5S and W1W sensors had stronger responses to the volatiles of melon samples, indicating that fresh oriental melon might have higher abundances of nitrogen oxides and terpene compounds. Especially, the response values of the W5S sensor were 1.3–2.0 times higher in all three forchlorfenuron application groups (10 mg/kg, 15 mg/kg, and 20 mg/kg) than in the pollination group (CK), suggesting that oriental melon in forchlorfenuron application groups may have high abundances of nitrogen oxides. For strawberries and avocados, more nitric oxide was detected in unripe fruits [29]. Previous studies reported that applying forchlorfenuron can result in a prolonged ripening process and delayed fruit maturity [30,31], which may be one reason for the higher concentrations of nitrogen oxides being detected in forchlorfenuron application groups.

Based on the responses of the E-nose sensors, the LDA method was used to reduce the differences within the classification and expand the differences between different groups in this study. As shown in Figure 1B, the variance contribution rates of LD1 and LD2 were 96.25% and 2.28%, respectively. Moreover, the LDA analysis showed the variation of each group along the abscissa (LD1) with a trend. The distance between the four groups was relatively far, especially between the pollination group (CK) and the other three forchlorfenuron application groups (10 mg/kg, 15 mg/kg, and 20 mg/kg), indicating that the three forchlorfenuron application groups were significantly different from the pollination group. These findings show that the E-nose could be applied to monitor the changes of volatile compounds in fresh oriental melon after forchlorfenuron application. However, none of the sensors of the E-nose were sensitive to esters in fresh oriental melon, which are the important volatile compounds of oriental melon. Therefore, more precise instruments (i.e., GC-IMS) were used in the subsequent experiments.

### 3.2. Qualitative Analysis of the Volatile Compounds by HS-GC-IMS

Figure 2 shows the three-dimensional (Figure 2A) and two-dimensional (Figure 2B) spectra obtained by HS-GC-IMS relying on chemical morphology. Different colors indicate different concentrations of the individual compounds, with white dots indicating a lower concentration and red dots indicating a higher concentration (Tian et al., 2020). Several single compounds might produce multiple signals or spots (dimers or even trimers), which are attributed to their varying concentrations [26].

It could be seen that volatile compounds were effectively separated from signal dots. Most of the signals appeared in the retention time of 100–700 s and the drift time of 1.0–2.0. A total of 48 volatiles were identified from the GC-IMS library (Table 1), including esters (27), alcohols (7), aldehydes (7), ketones (4), sulfides (2), and terpenes (1). These volatiles’ primary descriptive odor dimensions are fruity and ethereal, which have high odor strength. Among the 48 volatiles, esters, including 9 acetates and 18 nonacetate esters, were dominant quantitatively, accounting for 56.25%. The other quantity-predominant compounds were alcohols (14.58%), aldehydes (14.58%), and ketones (8.33%); sulfides and terpenes together accounted for only 6.25%. Consistent with the previous study, ethyl hexanoate, ethyl 2-methylbutyrate, and ethyl butanoate were considered key odorants in various melon fruits, having fruity, floral, and sweet odor [4,12]. Alcohols and aldehydes with nine carbon atoms, dominated by (Z)-non-6-enal, (E)-2-nonenal, and (3Z,6Z)-nona-3,6-dien-1-ol, which smelled “cantaloupe-like, cucumber-like”, were identified by many researchers as the characteristic components of the family Cucurbitaceae [4,11,12]. In this study, only n-Nonanal with nine carbon atoms was identified in oriental melon fruits. These differences could be attributed to geographical and cultivar variations as well as the different aroma extraction methods used. Compared with the oriental melon, the muskmelon has a different distribution profile of volatile compounds as has been reported, with aldehydes being the dominant compounds (33.33%), followed by esters (27.45%), alcohols (25.49%), and ketones (13.76%) [32].

### 3.3. Different Profiles of Volatile Flavor Compounds in Fresh Oriental Melon after Forchlorfenuron Application by HS-GC-IMS

To better understand the effect of volatile compounds in fresh oriental melon after forchlorfenuron application, the difference comparison model was applied to compare the differences between the different treatment groups. The topographic plot of fresh oriental melon from the pollination group (CK) was selected as a reference, and the spectrum of the other samples deducted the reference. If the volatile compounds were consistent, the background after deduction was white; red and blue indicated that the concentration of volatile compounds was higher and lower than the reference, respectively. As shown in Figure 3A, the number of red dots increased gradually with the increase in concentration of the forchlorfenuron application (10–20 mg/kg). Moreover, more red spots were observed for the mid-dose (15 mg/kg) and high-dose (20 mg/kg) groups. Further, several blue spots can be observed in the three forchlorfenuron application groups, indicating that the concentration of several volatile compounds in the three forchlorfenuron application groups was lower than in the pollination group (CK). 

The fingerprint was used to make an accurate judgment regarding the dense material on the topographic plot. As shown in Figure 3B,C, the signal intensities of methyl 2-methyl butanoate, amyl acetate, ethyl propanoate, (E)-hept-2-enal, methyl hexanoate, methyl isobutyrate, and dimethyl trisulfide were found to be higher in three forchlorfenuron application groups but were lower in the pollinated group (CK). These volatile compounds impart sweet, fruity, and sulfureous flavor. On the contrary, 2-methylbutanol imparts an alcoholic flavor, and 2-butanone possesses an ethereal fruity odor, which were significantly lower in the three forchlorfenuron application groups. A previous study showed that forchlorfenuron application could significantly influence the volatile flavor compounds in muskmelon [14]. The relative abundances of 14 volatile compounds emitted by the forchlorfenuron-treated fruits declined, including six ethyl esters, four aldehydes, three alcohols, and one ketone. By contrast, the relative abundance of 1-hexanol was on average higher in all the forchlorfenuron-treated fruits [11].

### 3.4. PCA and OPLS-DA Analysis

To better visualize the variability of volatile flavor compounds in fresh oriental melon after forchlorfenuron application, multivariate analyses (PCA and OPLS-DA) were attempted in this study. As shown in Figure 4A, all volatile compounds obtained from HS-GC-IMS were subjected to PCA analysis, with PC1 and PC2 accounting for 61% of the total variance. All samples from the pollination group (CK) are positioned on the left side of the score plot, and all melon samples from the forchlorfenuron application groups (10 mg/kg, 15 mg/kg, and 20 mg/kg) are positioned on the right side. Moreover, the proximity of the forchlorfenuron application groups indicates they have similar volatile compounds. The analysis results were consistent with those of the E-nose.

Supervised OPLS-DA was then conducted on the volatile compounds of fresh oriental melon samples to test the validity of PCA clustering and to further clarify the critical different aroma-active compounds of oriental melon after forchlorfenuron application. As shown in Figure 4B, a clear discrimination was achieved between the fresh oriental melon samples from different treatment groups. The OPLS-DA model explained a cumulative 62% of the total variance with high-quality performance parameters (R2Y = 0.874, Q2 = 0.894, and CV-ANOVA *p*-value < 0.05) [33]. The results indicate the goodness-of-fit and predictability of the PLS-DA model and confirm the excellent performance of GC-IMS combined with PLS-DA for the discrimination of oriental melon after forchlorfenuron application. Meanwhile, the VIP method of PLS-DA was used to screen the critical different aroma-active compounds of oriental melon after forchlorfenuron application. As shown in Table 2, the VIP scores of 13 volatile compounds were more than 1, indicating that these compounds might be used as critical markers in determining discrimination in the HS-GC-IMS PLS-DA model.

The 13 difference markers contain 12 esters, imparting a sweet fruity flavor in fresh oriental melon. Except for ethyl acetate and isobutyl acetate, the signal intensities of the other 10 esters in the pollination group (CK) were much lower than that of the three forchlorfenuron application groups (10 mg/kg, 15 mg/kg, and 20 mg/kg). Among them, ethyl 2-methyl butanoate was decreased appreciably in overripe cantaloupe as has been reported [34]. Consistent with the result of E-nose, applying forchlorfenuron could delay the fruit maturity of oriental melon [30], which may be one of the reasons for the decrease of these esters in the pollination group (CK).

The major limitation of this work includes the lack of data regarding the specific contribution of each compound to the aroma of fresh oriental fruits. Further studies are being designed to identify key aroma-active compounds in fresh oriental melon. The concentration and odor activity value (OAV) of critical volatile flavor compounds will be further calculated to verify the obtained results from this study. Furthermore, the mechanism of the changes in the types and concentrations of volatile compounds in oriental melon fruit after forchlorfenuron application is still unknown, and this needs further study.

## 4. Conclusions

In this study, the influence of forchlorfenuron on the volatile flavor compounds in fresh oriental melon was studied. E-nose can effectively separate the oriental melon from different application groups. The volatile profile of fresh oriental melon was characterized by establishing the fingerprint with GC-IMS. Forty-eight volatile components, including esters, alcohols, aldehydes, and ketones, along with smaller quantities of sulfides and terpenes, were identified from flesh oriental melon. Esters are the main substance of volatile compounds in fresh oriental melon. Compared to pollination melon fruits, the application of forchlorfenuron could significantly influence the content of esters in fresh oriental melon, which may be related to the prolonged ripening process of oriental melon after forchlorfenuron application. This work could be conducive to a better understanding of the characteristic aroma differences of fresh oriental melon after forchlorfenuron application and provide new theoretical guidance for the more rational use of forchlorfenuron.

## Figures and Tables

**Figure 1 foods-12-01272-f001:**
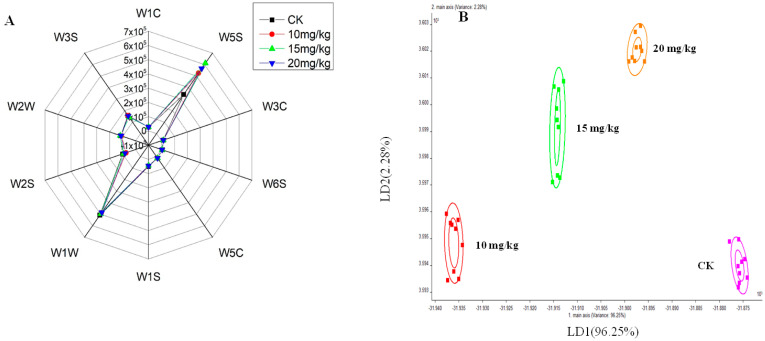
(**A**) Radar chart and (**B**) linear discriminant analysis (LDA) of fresh oriental melon after forchlorfenuron application obtained by E-nose measurement. CK: pollination group; 10 mg/kg: low-dose forchlorfenuron application group; 15 mg/kg: mid-dose forchlorfenuron application group; 20 mg/kg: high-dose forchlorfenuron application group.

**Figure 2 foods-12-01272-f002:**
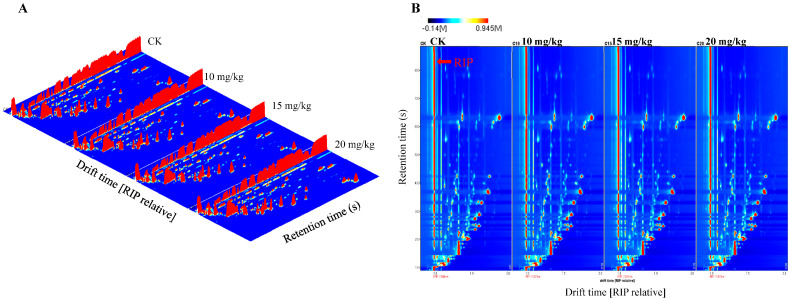
Volatile flavor compounds in fresh oriental melon after pollination or forchlorfenuron application: (**A**) a three-dimensional spectrum of the HS-GC-IMS response data; (**B**) a two-dimensional spectrum of the HS-GC-IMS response data.

**Figure 3 foods-12-01272-f003:**
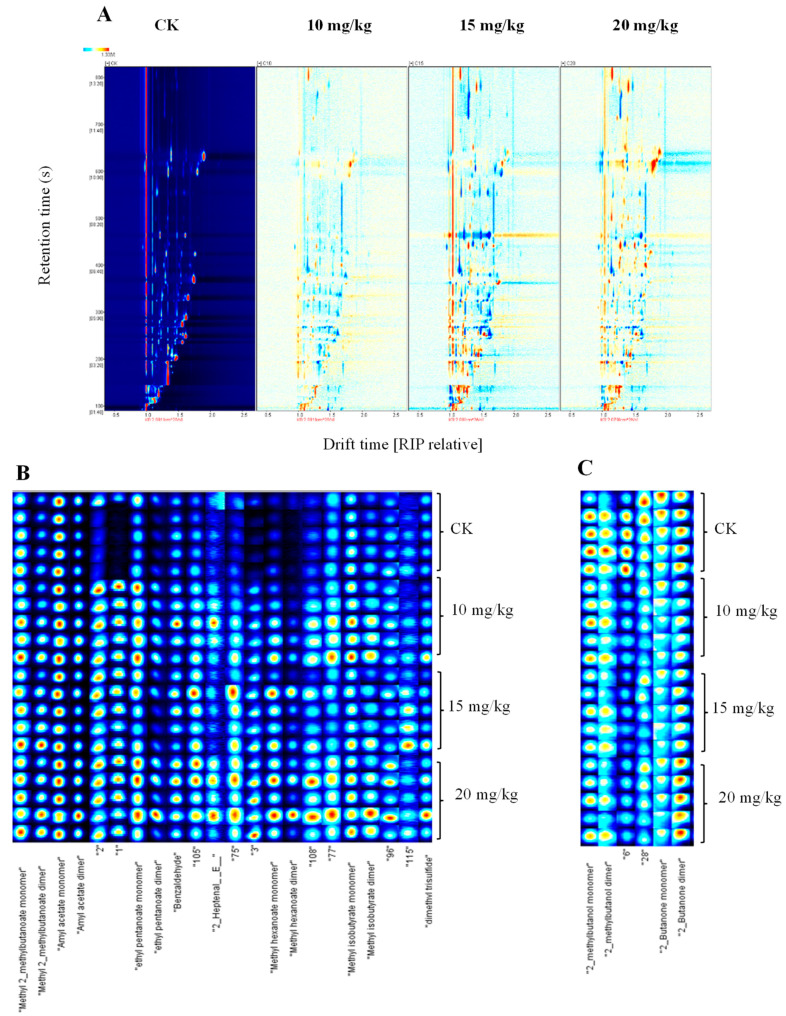
Differences of volatile flavor compounds of fresh oriental melon after pollination or forchlorfenuron application: (**A**) two-dimensional spectrum of the HS-GC-IMS response data; (**B**,**C**) gallery plot of the HS-GC-IMS response data.

**Figure 4 foods-12-01272-f004:**
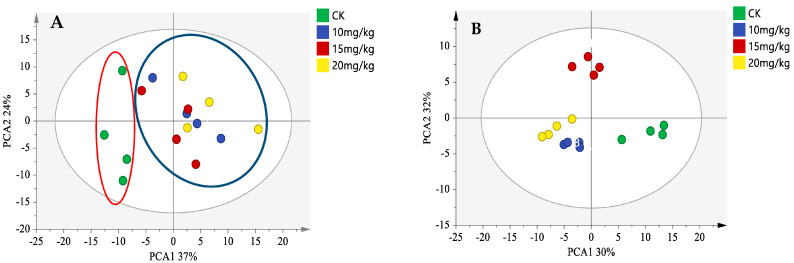
Score plots of the PCA model (**A**) and OPLS-DA mode (**B**) of volatile flavor compounds of fresh oriental melon after pollination or forchlorfenuron application.

**Table 1 foods-12-01272-t001:** Volatile flavor compounds in oriental melon identified by HS-GC-IMS.

Count	Compound	CAS#	Formula	MW	RI	Rt [sec]	Dt [RIPrel]	Comment	Odor Type	Odor Strength
Esters (27)										
1	Ethyl hexanoate	123-66-0	C_8_H_16_O_2_	144.2	1007.3	598.713	13.415	monomer	fruity	high
2	Ethyl hexanoate	123-66-0	C_8_H_16_O_2_	144.2	1007.3	598.713	17.987	dimer	fruity	high
3	Amyl acetate	628-63-7	C_7_H_14_O_2_	130.2	916.2	424.211	13.134	monomer	fruity	/
4	Amyl acetate	628-63-7	C_7_H_14_O_2_	130.2	916.2	424.211	17.643	dimer	fruity	/
5	3-Methylbutyl acetate	123-92-2	C_7_H_14_O_2_	130.2	880.9	368.989	12.962	monomer	fruity	high
6	3-Methylbutyl acetate	123-92-2	C_7_H_14_O_2_	130.2	882.4	371.198	17.425	dimer	fruity	high
7	Ethyl 2-methylbutanoate	7452-79-1	C_7_H_14_O_2_	130.2	852.4	332.543	12.448	monomer	fruity	medium
8	Ethyl 2-methylbutanoate	7452-79-1	C_7_H_14_O_2_	130.2	850.6	330.334	16.536	dimer	fruity	medium
9	Butyl acetate	123-86-4	C_6_H_12_O_2_	116.2	810.3	287.261	1237	monomer	ethereal	high
10	Butyl acetate	123-86-4	C_6_H_12_O_2_	116.2	813.7	290.574	16.208	dimer	ethereal	high
11	Ethyl butanoate	105-54-4	C_6_H_12_O_2_	116.2	796.7	274.007	12.058	monomer	fruity	high
12	Ethyl butanoate	105-54-4	C_6_H_12_O_2_	116.2	796.7	274.007	15.615	dimer	fruity	high
13	Ethyl 2-methylpropanoate	97-62-1	C_6_H_12_O_2_	116.2	756.0	236.253	11.927	monomer	fruity	high
14	Ethyl 2-methylpropanoate	97-62-1	C_6_H_12_O_2_	116.2	755.3	235.641	15.619	dimer	fruity	high
15	Ethyl propanoate	105-37-3	C_5_H_10_O_2_	102.1	709.0	198.307	1148	monomer	fruity	high
16	Ethyl propanoate	105-37-3	C_5_H_10_O_2_	102.1	709.0	198.307	14.528	dimer	fruity	high
17	Methyl 2-methylbutanoate	868-57-5	C_6_H_12_O_2_	116.2	776.1	254.614	11.927	monomer	fruity	/
18	Methyl 2-methylbutanoate	868-57-5	C_6_H_12_O_2_	116.2	774.8	253.39	1533	dimer	fruity	/
19	Ethyl Acetate	141-78-6	C_4_H_8_O_2_	88.1	590.9	145.231	10.968	monomer	ethereal	high
20	Ethyl Acetate	141-78-6	C_4_H_8_O_2_	88.1	599.6	148.029	13.349	dimer	ethereal	high
21	Methyl isobutyrate	547-63-7	C_5_H_10_O_2_	102.1	687.5	184.409	11.419	monomer	fruity	/
22	Methyl isobutyrate	547-63-7	C_5_H_10_O_2_	102.1	688.1	184.759	14.424	dimer	fruity	/
23	Isobutyl acetate	110-19-0	C_6_H_12_O_2_	116.2	767.5	246.674	16.135		fruity	medium
24	Ethyl pentanoate	539-82-2	C_7_H_14_O_2_	130.2	901.4	399.717	12.764	monomer	fruity	high
25	Ethyl pentanoate	539-82-2	C_7_H_14_O_2_	130.2	901.4	399.717	16.829	dimer	fruity	high
26	Methyl hexanoate	106-70-7	C_7_H_14_O_2_	130.2	925.4	440.483	12.895	monomer	fruity	medium
27	Methyl hexanoate	106-70-7	C_7_H_14_O_2_	130.2	926.1	441.667	16.845	dimer	fruity	medium
Alcohols (7)										
28	Ethanol	64-17-5	C_2_H_6_O	46.1	483.8	110.843	10.485	monomer	alcoholic	medium
29	Ethanol	64-17-5	C_2_H_6_O	46.1	484.8	111.146	11.285	dimer	alcoholic	medium
33	1-Octen-3-ol	3391-86-4	C_8_H_16_O	128.2	985.1	555.176	11.583	monomer	earthy	high
31	1-Octene-3-ol	3391-86-4	C_8_H_16_O	128.2	986.2	557.384	15.954	dimer	earthy	high
32	1-Octen-3-ol	3391-86-4	C_8_H_16_O	128.2	985.1	555.176	17.302	polymer	earthy	high
33	2-Methylbutanol	137-32-6	C_5_H_12_O	88.1	736.3	219.252	12.326	monomer	ethereal	medium
34	2-Methylbutanol	137-32-6	C_5_H_12_O	88.1	738.5	221.168	1472	dimer	ethereal	medium
Aldehydes (7)										
35	Benzaldehyde	100-52-7	C_7_H_6_O	106.1	957.5	500.837	11.488		fruity	high
36	3-Methylbutanal	590-86-3	C_5_H_10_O	86.1	665.8	173.049	11.584	monomer	aldehydic	high
37	3-Methylbutanal	590-86-3	C_5_H_10_O	86.1	661.3	170.97	14.065	dimer	aldehydic	high
38	n-Nonanal	124-19-6	C_9_H_18_O	142.2	1106.9	783.669	14.789	monomer	aldehydic	high
39	n-Nonanal	124-19-6	C_9_H_18_O	142.2	1106.9	783.669	19.299	dimer	aldehydic	high
40	(E)-Hept-2-enal	18829-55-5	C_7_H_12_O	112.2	956.9	499.653	16.611		green	high
41	Pentanal	110-62-3	C_5_H_10_O	86.1	697.5	190.514	1183		fermented	/
Ketones (4)										
42	Acetone	67-64-1	C_3_H_6_O	58.1	500.2	116.093	11.249		solvent	high
43	2,3-Butanedione	431-03-8	C_4_H_6_O_2_	86.1	582.4	142.503	11.657		buttery	high
44	2-Butanone	78-93-3	C_4_H_8_O	72.1	584.4	143.132	10.575	monomer	ethereal	/
45	2-Butanone	78-93-3	C_4_H_8_O	72.1	583.3	142.782	12.482	dimer	ethereal	/
Sulfides (2)										
46	Butyl sulfide	544-40-1	C_8_H_18_S	146.3	1072.9	721.533	12.851		alliaceous	high
47	Dimethyl trisulfide	3658-80-8	C_2_H_6_S_3_	126.3	947.6	481.593	13.084		alliaceous	/
Terpenes (1)										
48	Alpha-Pinene	80-56-8	C_10_H_16_	136.2	938.3	464.062	12.175		herbal	high

The odor type and odor strength were obtained from The Good Scents Company Information System.

**Table 2 foods-12-01272-t002:** The 13 difference markers of the volatile flavor compounds of oriental melon under different treatment groups.

Compound	CAS#	Formula	Odor Descriptor	Peak Area			
				CK	10 mg/kg	15 mg/kg	20 mg/kg
Esters							
Ethyl acetate dimer	141-78-6	C_4_H_8_O_2_	ethereal fruity sweet weedy green	33,030.65	31,044.70	31,555.70	31,832.88
3_Methylbutyl acetate dimer	123-92-2	C_7_H_14_O_2_	sweet fruity banana solvent	9685.77	10,484.30	10,242.62	10,906.42
Ethyl propanoate dimer	105-37-3	C_5_H_10_O_2_	sweet fruity rum juicy fruit grape pineapple	9154.11	9455.36	9616.75	9970.14
Isobutyl acetate	110-19-0	C_6_H_12_O_2_	sweet fruity ethereal banana tropical	5429.39	4908.41	4740.68	4901.22
Ethyl hexanoate dimer	123-66-0	C_8_H_16_O_2_	sweet fruity pineapple waxy green banana	1320.87	3116.89	2774.75	3979.39
Ethyl hexanoate monome	123-66-0	C_8_H_16_O_2_	sweet fruity pineapple waxy green banana	2109.47	3067.76	2806.68	3155.46
Amyl acetate dime	628-63-7	C_7_H_14_O_2_	ethereal fruity banana pear banana apple	1403.12	2094.79	1673.21	2559.41
Ethyl 2-methylbutanoate dime	7452-79-1	C_7_H_14_O_2_	sharp sweet green apple fruity	6161.62	6653.40	6602.08	7472.25
Ethyl 2-methylbutanoate monome	7452-79-1	C_7_H_14_O_2_	sharp sweet green apple fruity	1103.21	1326.94	1325.19	1380.20
Ethyl pentanoate monomer	539-82-2	C_7_H_14_O_2_	sweet fruity apple pineapple green tropical	492.25	889.20	661.18	972.12
Methyl 2-methylbutanoate dime	868-57-5	C_6_H_12_O_2_	ethereal estery fruity tutti frutti green apple lily of the valley powdery fatty	491.19	679.60	770.23	900.93
Methyl hexanoate monomer	106-70-7	C_7_H_14_O_2_	fruity pineapple ether	137.25	379.01	441.57	604.47
Aldehydes							
Benzaldehyde	100-52-7	C_7_H_6_O	strong sharp sweet bitter almond cherry	241.09	389.94	335.86	425.24

## Data Availability

The data that support the findings of this study are available from the corresponding author upon request.

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
