# Peer review of "Discrimination and Characterization of Volatile Flavor Compounds in Fresh Oriental Melon after Forchlorfenuron Application Using Electronic Nose (E-Nose) and Headspace-Gas Chromatography-Ion Mobility Spectrometry (HS-GC-IMS)"

_foods, 2023, doi:10.3390/foods12061272_

Round 1

Reviewer 1 Report

This manuscript contains novel information of great interest to the field of study. However, there are many mistakes that should be correct. The manuscript presents flaws in the experimental design, the statistics should be improved, and flaws have also been detected in the references of the cited articles. The aim of the study is not clear.

Author Response

Response to Reviewer 1 Comments

Point 1: This manuscript contains novel information of great interest to the field of study. However, there are many mistakes that should be correct. The manuscript presents flaws in the experimental design, the statistics should be improved, and flaws have also been detected in the references of the cited articles. The aim of the study is not clear.

Response 1: Thank you very much. According to your suggestion and other reviewers’ suggestions, we have corrected errors throughout the manuscript. The experimental design and statistics also improved in the revised manuscript as follows:

“2.1. The Oriental melon Field trials.

Oriental melons (Cucumis melo var. makuwa) were cultivated in a greenhouse at the Chinese Academy of Agricultural Sciences (Beijing, China), from July to October 2018. The temperature of the greenhouse was maintained in the range of 25–30 °C, with 60% of average relative humidity throughout the experiment. The field trials were divided into four groups: three forchlorfenuron application groups and one pollination group. In the treatment groups, doses of forchlorfenuron soluble concentrate (SL) were set from 10 to 20 mg/L according to the recommended dose in the registered label; and the melon ovary was completely dipped with 10 mg/L, 15 mg/L, 20 mg/L forchlorfenuron solutions for 1-2 s, respectively; In the pollination group, only chasmogamy of female flowers by male flowers. All fruit set treatments were performed on the same morning (6–9 AM).

Representative melon fruit samples were harvested with the best edible quality according to the experience of melon farmers (34 days after pollination or forchlorfenuron application). Mature melon fruits were selected using a combination of different harvest indices, including smooth-skinned with sweet and fragrant pulp, aroma emission detected by the human nose, pale yellow skin color, and peduncle suberization (Figure S1).11 And melons were selected in this experiment based on uniform size, weight, and color, and collected at least six fruits in each group. Melon samples were placed in polyethylene bags and transported to the laboratory for the next stage.” (Lines 64-85 in page 2)

“Supervised OPLS-DA was then conducted on the volatile compounds of fresh oriental melon samples to test the validity of PCA clustering and to further clarify the critical different aroma-active compounds of oriental melon after forchlorfenuron application. As shown in Figure 4B, clear discrimination was achieved between the fresh oriental melon samples from different treatment groups. The OPLS-DA model explained a cumulative 62% of the total variance with high-quality performance parameters (R2Y = 0.874, Q2 = 0.894, and CV-ANOVA p-value < 0.05).33” (Lines 237-243 in page 9)

At the same time, the references of the cited articles were examined and updated in the revised manuscript.

The objective of the present study is to identify the differentiation of flavor changes in oriental melon treated with different concentrations of forchlorfenuron. E-nose and HS-GC-IMS were applied to characterize the volatile compound composition and content when the oriental melon fruits were harvested after maturation. The results will provide new theoretical guidance for more appropriate use of forchlorfenuron in oriental melon. According to your suggestion, we have also rewritten the introduction in the revised manuscript as follows:

“Oriental melon is a species of thin-pericarp melon,1 and it has the largest plantation in China, accounting for about 51% of the total global production. Oriental melons are often planted in the greenhouse to increase the price of fruits. However, the lack of pollinators often affects the fruit set rate for facilities. Forchlorphenuron is a synthetic cytokinin-like growth regulator, which can act synergistically with endogenous auxins to induce parthenocarpy and promote cell expansion.2 In recent years, forchlorphenuron has been extensively used in oriental melon cultivation to improve the fruit set.

With the increasing prevalence of forchlorfenuron application, more and more studies have focused on its influence on fruit quality.3,4 Several studies have shown that application of forchlorfenuron decreased sucrose and glucose content and increased bitterness in melon.5,6 In addition to sugar, volatile aroma plays a decisive role in purchasing oriental melon.7 The volatile components of melon have been analyzed in previous reports and approximately 300 compounds have been identified.8-11 They produce volatile aldehydes, alcohols, and especially large quantities of esters, likely to be the key contributors to their unique aroma.12-14 However, limited studies have reported the effect of forchlorfenuron on the aroma compounds of oriental melon fruit. Although Li et al found that the abundance of volatile compounds was decreased after forchlorfenuron application in muskmelon using a gas chromatography-mass spectrometer,13 the findings have generally been obtained from frozen samples alternative to fresh samples. It has been reported that significant changes in volatiles occurred during the freezing process in fruits and vegetables.15-17 To accurately evaluate the effect of forchlorfenuron on aroma characteristics, a quick and straightforward method to discriminate the variation of aroma volatiles using fresh oriental melon fruit is critical.

Sensory analysis using trained panelists has been employed conventionally to evaluate the variation of fruit aroma, which can directly measure the fruit flavor intensity. However, this method is expensive and time-consuming, with low objectivity and reproducibility.18,19 Electronic nose (E-nose) and gas chromatography–ion mobility spectrometry (GC-IMS) as emerging and cost-effective techniques for volatile compounds analysis offer advantages of fast analysis timeframes, high sensitivity, low detection limits, and simple sample pretreatment. 20-24 These techniques have been successfully utilized individually or in combination in many fields, mainly involving freshness prediction, adulteration identification, and food composition classification. Ezhilan et al. discriminated the pathogen contamination of apples using E-nose; higher classification accuracy was attained with an accuracy of 99.9%.25 Guo et al. study showed the potential of GC-IMS-based approaches to evaluate the volatile compound profiles of fresh-cut yam at different stages in the yellowing period. 26

Therefore, the present study aims to identify the differentiation of flavor changes in oriental melon treated with different concentrations of forchlorfenuron. E-nose and HS-GC-IMS were applied to characterize the volatile compound composition and content when the oriental melon fruits were harvested after maturation. The results will provide new theoretical guidance for more appropriate use of forchlorfenuron in oriental melon.” (Lines 25-65 in page 1)

Reviewer 2 Report

The paper can be revisions:

1-      The introduction needs improvement with recently published literature review. The purpose of the research is not well reported in the final part of the introduction. The motivation of the paper is not sufficiently justified in the introduction.

2-      Including a photograph is recommended that gives certainty that the measurement systems was used.

3-      Provide a real view of the samples used in the study.

4-      The paper needs more clarification. The contribution of the study is not clear? What is precisely proposed in the study, and what is the actual contribution to the literature? It should be explained clearly. It would be excellent if the importance of this issue was validated by detailed research and thoroughly documented data.

5-      The Limitations of the proposed study need to be discussed before conclusion.

6-      The findings in "Conclusions" Section should be stated point by point. A contextualization has to be added as incipit, in order to make the Conclusions section self-standing. Please make a few-line conclusion about the work. It can be the essence of all the results. It should be suggestive about the best practice for future works.

7-      Author should add separate section regarding future outlook and specific comment point wise based on their study.

8-      Some grammatical errors and typos should be corrected.

Author Response

We sincerely adopted   your comments and revised the paper according to your suggestions. The points of the revision were detailed in the attached letter.

We are looking forward to your kind suggestions for the publication of this paper.

Reviewer 3 Report

Comments to the authors

The authors aimed to discriminate the volatile profile of pollinated and forchlorfenuron treated fresh oriental melon by E-nose and GC-IMS techniques. The manuscript seems interesting also due to the use of chemometrics to investigate the differences of the different volatile profiles. In my opinion some revisions are required:

In general, the introduction does not clearly express the research background, many things are not relevant to the topic, speaking in general of the instrumental methods. Please add some references on the matrix studied.

L37-38 please rephrase the sentence

L43 The comma between “12, 13” must be superscript  

L58-64 it is a sort of conclusion, rephrase into a more general way

L93 which is the carrier gas? please specify

Section 2.4 two column were used to separate the compounds ?

L140 The comma between “25, 26” must be superscript  

L167 delete the dot after the word “Figure”

L182 “Flow”?

L369 insert DOI

Author Response

(The authors gave the same response as above.)

Reviewer 4 Report

Manuscript ID: foods- 2118031

Title: Discrimination and characterization of volatile flavor compounds in fresh oriental melon after forchlorfenuron application using electronic nose (E-nose) and headspace-gas chromatography-ion mobility spectrometry (HS-GC-IMS)

Comments:

The manuscript is interesting and presents new information on the evaluation of the flavor of oriental melon after forchlorfenuron application. If forchlorfenuron application provides a sweet fruity flavor, I think a sensory test as a quantitative descriptive analysis would help to distinguish small differences in sensory attributes selected by a trained panel. Additionally, this would allow to know if the differences instrumentally reported can be detected by consumers.

The following comments should be taken into account to enrich the writing and give a more general approach.

Line 42. Please justify why freezing modifies the abundance of volatile compounds. Why did the authors use frozen and not fresh samples?

Line 47. Considering the olfactory-detection limit of human, and the cost and time of the analysis, could the authors please justify briefly the use of these technologies (E-nose and GC-IMS) rather than applying sensory analysis using trained panelists. Justify with bibliographical references.

Line 58. Why do you include the results of your research in the introduction? Please remove this part. Please justify more adequately your research.

Line 70. Justify the use of the concentrations of forchlorfenuron used.

Line 75. Harvest was carried out 34 days after pollination or forchlorfenuron application. However, the appropriate time for harvest is highly variable (24-45 days) depending on the temperature. Add a color table to know the ideal ripeness point chosen. Indicate if the fruits were harvested at commercial or physiological maturity.

Line 77. Indicate the range of size, weight and color of the samples, in order to ensure homogeneity in the experimental units.

Line 118. Why did you use multivariate techniques for data analysis? the use of analysis of variance could be more convenient for analysis of experimental data. Regarding the data, were the assumptions regarding normal distribution, independence and homogeneity of variance not met? Justify.

Line 139. Do the fruits were harvested at different ripeness points?

Line 162. Hence, the importance of complementing the study with sensory analysis techniques with a trained panel.

Author Response

(The authors gave the same response as above.)

Round 2

Reviewer 2 Report

The changes made were satisfactory. Hence the manuscript can be accepted.

Reviewer 4 Report

The manuscript has been improved and I appreciate the authors' efforts to respond my comments. The introduction and materials and methods has been reinforced according my suggestions.